# Diversified Stimuli-Induced Inflammatory Pathways Cause Skin Pigmentation

**DOI:** 10.3390/ijms22083970

**Published:** 2021-04-12

**Authors:** Md Razib Hossain, Tuba M. Ansary, Mayumi Komine, Mamitaro Ohtsuki

**Affiliations:** Department of Dermatology, Faculty of Medicine, Jichi Medical University, Tochigi 329-0498, Japan; razib@jichi.ac.jp (M.R.H.); tuba2020@jichi.ac.jp (T.M.A.); mamitaro@jichi.ac.jp (M.O.)

**Keywords:** melanosome, melanocytes, melanogenesis, skin pigmentation, inflammation, inflammatory cytokine

## Abstract

The production of melanin pigments by melanocytes and their quantity, quality, and distribution play a decisive role in determining human skin, eye, and hair color, and protect the skin from adverse effects of ultraviolet radiation (UVR) and oxidative stress from various environmental pollutants. Melanocytes reside in the basal layer of the interfollicular epidermis and are compensated by melanocyte stem cells in the follicular bulge area. Various stimuli such as eczema, microbial infection, ultraviolet light exposure, mechanical injury, and aging provoke skin inflammation. These acute or chronic inflammatory responses cause inflammatory cytokine production from epidermal keratinocytes as well as dermal fibroblasts and other cells, which in turn stimulate melanocytes, often resulting in skin pigmentation. It is confirmed by some recent studies that several interleukins (ILs) and other inflammatory mediators modulate the proliferation and differentiation of human epidermal melanocytes and also promote or inhibit expression of melanogenesis-related gene expression directly or indirectly, thereby participating in regulation of skin pigmentation. Understanding of mechanisms of skin pigmentation due to inflammation helps to elucidate the relationship between inflammation and skin pigmentation regulation and can guide development of new therapeutic pathways for treating pigmented dermatosis. This review covers the mechanistic aspects of skin pigmentation caused by inflammation.

## 1. Introduction

Skin color is one of the topmost concerns for human beings as it provides uniformity, identity and represents the overall appearance of a person [1]. Production, quantity, quality, and distribution of melanin, a group of natural pigments, determines the color of human skin, eyes, and hair. It is crucially involved in the defense mechanism to protect skin from adverse effects of ultraviolet radiation (UVR) and oxidative stress of environmental pollutants [2]. Synthesis of melanin starts with formation of melanosomes, melanin-containing organelles, in epidermal melanocytes by a process called melanogenesis [3,4]. Melanocytes are specialized cells derived from unpigmented precursor cells called melanoblasts, originating from embryonic neural crest cells which can migrate towards the skin and other tissues during embryogenesis [5,6,7,8]. Skin pigmentation is a specific and complex mechanism that occurs due to accumulation of melanosomes in keratinocytes to protect skin from solar irradiation. Melanin pigment can be divided into two types—pheomelanin and eumelanin. Pheomelanin has a reddish to brownish color, causing reactive oxygen species production under UVR stimulation. Eumelanin has black color, which works mainly to protect the nucleus from UVR. Skin color differences can emerge from the quantity and quality (pheo/eumelanin ratio) of melanin produced as well as size, number, composition, mode of transfer, distribution, and degradation of melanosomes inside keratinocytes, whereas melanocytes numbers typically remain relatively constant [8]. Keratinocytes have a significant role in regulating adhesion, proliferation, survival, and morphology of melanocytes [8].

There are various nonspecific intrinsic factors (hormonal environment, inflammation) and extrinsic factors (solar irradiation such as ultraviolet irradiation, environmental pollution, drugs) modulating genetically determined melanin levels. Various stimuli including skin eruptions such as eczema, allergens, microbial infection, pathogens, chemical stimuli and physical damage, ultraviolet light exposure, and aging can all lead to skin inflammation [9,10,11,12], defined as a defensive reaction of living tissues with a vascular system in response to exogenous and endogenous stimuli [13]. In dermatology and cosmetology, hyperpigmentation or hypopigmentation after inflammation is a major problem and commonly seen due to acute or chronic inflammatory skin reactions [14]. Inflammatory mediators are chemical factors involved in mediating inflammatory reactions and mainly secreted by T helper cells (Th), monocytes and macrophages, dendritic cells, as well as epidermal keratinocytes. Recent studies have revealed a close relationship between inflammatory cytokines and skin pigmentation. Diversified interleukins (ILs) and varieties of inflammatory mediators can also participate in melanogenesis regulation by modulating proliferation and differentiation of human epidermal melanocytes, and also promote or inhibit melanogenesis-related gene expression directly or indirectly [14,15,16]. Many genes are involved in regulating pigmentation at various levels, as well as mutations in several cause pigmentary disorders.

The above studies draw attention to the relationship between inflammation and skin pigmentation regulation, as well as guide new therapeutic pathway development for treating pigmented dermatosis. In this review, we describe the mechanistic perspectives of skin pigmentation due to inflammation.

## 2. Melanogenesis Process

Skin epidermal units are responsible for melanin production by a complex multistage process called melanogenesis. These units are composed of melanocytes surrounded by keratinocytes. Melanocyte interacts with the endocrine, immune, inflammatory and central nervous systems, and its activity is also regulated by extrinsic factors such as UVR and drugs. Melanocytes produce melanin pigment in melanosomes, which are transferred to surrounding keratinocytes. The production of melanin in melanocytes and accumulation of melanin-containing melanosomes in keratinocytes leads to skin pigmentation and its dysregulation causes different types of pigmentation defects, which are classified as hypopigmentation (including oculocutaneous albinism (OCA)), hyperpigmentation (such as melasma) or mixed hyper-/hypopigmentation (including dyschromatosis symmetrica hereditaria) and may occur with or without alteration of the number of melanocytes [8,17,18]. Congenital defects in melanin synthesis or melanosome function or migration of melanoblasts cause congenital hypopigmented disorders, such as OCA, Chediak Higashi syndrome, or piebaldism. On the other hand, congenital hyperpigmented conditions depend more on locally distributed melanocytes or nevus cells, other types of melanin-producing cells derived from neural crest cells such as linear and whorled nevoid hypermelanosis, and congenital melanocytic nevus. Acquired conditions include vitiligo vulgaris, halo nevus, melasma, and senile pigmented macules, which mainly depend on immune dysfunction, inflammation or aging.

Ectodermal neural crest melanoblasts are the origin of melanocytes, and melanoblasts migrate to different destinations as the embryo develops, including the basal layer of the epidermis, hair follicles, eye uvea, vestibular organ, endolymphatic sac and pia meter [19,20]. Various mediators and transcription factors are involved in maturation and migration of melanoblasts to their final destination, such as Paired box protein Pax-3 (PAX3), sex-determining region Y-box 10 (SOX10), endothelin (EDN) 3, Microphthalmia-associated transcription factor (MITF), and stem cell growth factor receptor (KIT) and its ligand stem cell growth factor (SCF) [20,21]. Once they differentiate into melanocytes in the epidermis, various factors produced by the surrounding keratinocytes influence the function of melanocytes, such as SCF, EDN1, Nerve growth factor (NGF), Hepatocyte growth factor (HGF), granulocyte-macrophage colony stimulating factor (GM-CSF), α-Melanocyte-stimulating hormone (α-MSH), Adrenocorticotropic hormone (ACTH), prostaglandin F2 (PGF2) and prostaglandin E2 (PGE2). 

It is postulated that solar lentigines (SLs), hyper-pigmentary spots on sun-exposed skin of aged persons, could be treated by combined topical treatment with EDN signaling blockers and tyrosinase inhibitors [22], and SCF inhibitors or EDN1-triggered intracellular signaling cascades can downregulate the hyperpigmentation observed in solar lentigo, UVB (ultraviolet ray-B) induced melanosis, and melasma [23]. Melanosomes, subcellular lysosome-like organelles in melanocytes, store melanin [24,25] and synthesize two types of melanin pigment: eumelanin (brown-black or dark insoluble polymer) and pheomelanin (red-yellow soluble polymer). After synthesis, mature melanosomes are carried by molecular motors such as kinesins from the perinuclear region to dendrites of melanocytes, interacting with microtubules. On the other hand, dynein works in maintaining perinuclear distribution of melanosomes in melanocytes [26,27]. In the epidermis, each melanocyte is surrounded by 30 to 40 keratinocytes through dendrites and transfers mature melanosomes into the cytoplasm of keratinocytes [8,19]. The size and the number of melanosomes, as well as amount, type, transfer, and distribution of melanin in keratinocytes, determine phenotypic diversity of pigmentation, whereas melanocyte numbers are relatively constant in different ethnic groups [8,19,20]. Delayed degradation of melanosomes in keratinocytes has been found to increase visible pigmentation in dark-skinned individuals which is due to large, more numerous, and elongated melanosomes [19,20,28]. These differences in melanosomes are observed at the time of birth and are not determined by extrinsic factors such as UVR, which indicates that mutation in specific genes is related to the control of skin pigmentation.

Various molecules are involved in skin pigmentation, divided into three distinct groups: proteins required for melanosome structure, enzymes involved in melanin synthesis, and proteins required for melanosome transport and distribution [29]. There are four stages (I–IV) of melanosomal maturation and three important structural proteins (Figure 1) forming melanosomes, including melanosomal matrix protein (PMEL17/Silv/GP100), melanoma antigen recognized by T cell-1 (MART-1), and glycoprotein nonmetastatic melanoma protein b (GPNMB/DC-HIL/osteoactivin). Stages I and II are considered as pre-melanosomes and do not contain melanin. In stage III melanosomes, melanin is deposited on PMEL17 fibrils. Melanin fills the melanosome in the last stage and forms a masked internal structure and dark color. MART-1, found in earlier melanosomal stages, is essential for the maturation of PMEL17 [30]. GPNMB, a melanosome-specific and proteolytically released protein, is abundant in late melanosomes [31] and critical for the formation of melanosomes in an MITF-independent fashion [32]. Scaffold materials are provided by these structural melanosomal proteins for melanin synthesis.

Melanogenic enzymes need to be delivered in melanosomes to reach the final stage of melanosomal maturation and melanin synthesis (stages III–IV). There are three key enzymatic components of melanosomes playing central roles in melanin synthesis: tyrosinase (TYR), tyrosinase-related protein-1 (TYRP1), and dopachrome tautomerase/tyrosinase-related protein-2 (DCT/TYRP2). TYR is a copper-dependent enzyme catalyzing conversion of L-tyrosine into L-3,4-dihydroxyphenylalanine (L-DOPA), the rate-limiting stage in melanin synthesis. This enzyme is deactivated by copper oxidization but can be activated by electron donors such as L-DOPA, ascorbic acid, superoxide anion, and possibly nitric oxide (NO) [14,17,33]. Protein kinase C-β (PKC-β) phosphorylates two serine residues of the cytoplasmic domain which are also important for tyrosinase activation [34]. A most severe form of albinism OCAIa occurs due to the mutation inactivating this enzyme. TYRP1 and TYRP2 also exist in the membrane of melanosomes. Although its specific role is not yet well-established, it is assumed that TYRP1 has a role in activation and stabilization of tyrosinase, melanosome synthesis, increasing eumelanin/pheomelanin ratio and working against oxidative stress due to its peroxidase effect [14,17].

Melanosomes are transported from the perinuclear area to periphery of melanocytes, and melanocytes transfer packaged melanin into adjacent keratinocytes for skin pigmentation (Figure 1) [29]. Early melanosomes, originating in the perinuclear area, gradually become mature then turn to late (pigmented) melanosomes, while they move toward periphery of melanocytes (i.e., dendrites). Microtubules and collaboration of microtubule-associated motor proteins (kinesins and dyneins) are involved in trafficking of mature melanosomes from the melanocyte perinuclear area to dendrites [26,27]. Kinesins ensure centrifugal movement of melanosomes, and melanosomal cargo is transferred from microtubules to F-actin in dendrites via the complex of RAB27A, melanophilin (MLPH) and myosin-Va (MYO5A). It is demonstrated by mouse Melan-a cells that RAB27A links to synaptotagmin-like 2 (SYTL2) prompting SYTL2 to dock melanosome at the plasma membrane [35,36].

However, the exact transfer mechanism of human melanosome to keratinocytes is still unclear, but four different mechanisms have been proposed: (i) exocytosis, (ii) cytophagocytosis, (iii) fusion of plasma membrane, and (iv) transfer by membrane vesicles [37].

## 3. Melanogenesis Regulation

MITF is a basic helix–loop–helix leucine zipper (bHLH-ZIP) transcription factor belonging to the MIT regulating melanocyte function. MITF controls expression of the melanogenesis enzymes TYR, TYRP1 and TYRP2 [38] and regulates melanocyte functions, including pigmentation, melanocyte differentiation, proliferation, and cell survival [39]. Furthermore, MITF protein plays a vital role in expression regulation of the RAB27A protein, crucial in melanosome transport and the melanosomal matrix protein Pmel17 [17,21,40,41]. It has also been discovered that the MIT (microphthalmia) family is the main transcriptional regulator of many genes coding for proteins involved in pigment synthesis, melanosomal structure, as well as melanosome trafficking [38]. Many other transcription factors are involved in regulating MITF expression, including a paired box protein 3 (PAX3), sex-determining region Y-box 9 and 10 (SOX9, SOX10), lymphoid enhancer-binding factor 1 (LEF-1) and Cyclic adenosine monophosphate (cAMP) responsive-element binding protein (CREB), which is phosphorylated by signals via melanocortin-1 receptor (MC1R) [29]. Recent findings have elucidated MITF expression regulation in mice melanocytes is influenced by salt-inducible kinase (SIK), which phosphorylates CREB-regulated transcription coactivator (CRTC) to prevent its nuclear translocation, resulting in inhibited CREB-induced MITF upregulation [42].

### Involvement of Signaling Pathways in Melanogenesis Regulation

The regulation of melanogenesis is a complex process and more than 150 genes are involved in this regulation [43]. Melanogenesis can be controlled at different levels as many of these genes affect the developmental process crucial to melanoblasts, while others are involved in melanocyte survival and differentiation [44]. At least 25 of these genes are involved in regulating different pigmentation pathways or function of melanosomes. The cAMP/protein kinase A (PKA) signaling pathway is one of the most significant signaling pathways and the most well-known receptor is the MC1R. MC1R belongs to G protein-coupled receptor family and is expressed predominantly in melanocytes. The α-melanocyte-stimulating hormone (α-MSH) binds to MC1R and activates adenylate cyclase (AC), increasing intracellular concentration of secondary messenger, cAMP. The cAMP activates PKA, then PKA phosphorylates CREB, leading to increased expression of MITF, resulting in melanogenesis promotion [17,28,45,46].

In addition to the above, other signaling pathways such as mitogen-activated protein kinase (MAPK), inositol trisphosphate/diacylglycerol (IP3/DAG), wingless-type protein (WNT), and PKC are reported to participate in the melanogenesis process. The IP3/DAG pathway is activated by the α1 adrenergic receptor (α1AR) and thus increases the intracellular levels of PKC-β and activates tyrosinase [47]. The c-KIT, granulocyte-macrophage colony-stimulating factor receptor (GM-CSFR), and HGF receptor-mediated signaling pathways can be activated by SCF, GM-CSF and HGF, respectively, which leads to autophosphorylation and activation of MAPK, thereby phosphorylating MITF and upregulating melanogenesis-related enzyme expression [3,48,49]. The WNT bind to class frizzled (FZD) receptors and the wingless-type protein (WNT)/FZD signaling pathway is involved in several aspects of the mouse melanocyte physiology, from developmental lineage differentiation to subsequent maintenance of melanocytes [50]. Indeed, pigment development and differentiation from the neural crest cells into melanocytes can be promoted by WNT1 and WNT3A in zebrafish [51]. Several models including mouse Melan-a cells and avian embryo melanocytes have demonstrated WNT1 is responsible for increasing the number of melanoblasts differentiating from melanoblast precursors, whereas WNT3A is involved in promotion of the differentiation of neural crest cells and melanoblasts into melanocytes by maintaining MITF expression [52,53,54]. In general, the WNT signaling pathway can activate the melanocytes-specific MITF isoform (MITF-M) promoter [55,56,57] and upregulate MITF expression, thus participating in melanogenesis regulation. Catecholamines can promote melanogenesis through the cAMP/PKA pathway and also mediate melanogenesis through PKC-β pathways by α1 and β2 adrenergic receptors [3,47]. 

## 4. Skin Inflammation

The skin is the largest organ in the human body and provides the first line of defense against pathogens and other environmental pollutants [58]. These external stimuli as well as intrinsic stimuli, such as neurotransmitters, advanced glycation end products, amyloids, and debris of dead cells, provoke either short-term or long-term skin inflammation to the host [59]. Consequently, the production of reactive oxygen species (ROS), known as the first line of defense against pathogen and external pollutants, is promoted either directly or indirectly, and thus ROS stimulates the inflammation process [60,61]. External and intrinsic stimuli trigger inflammation, resulting in immunologic reactions (e.g., dendritic cell activation followed by T cell activation, and finally leading to inflammatory disorders, such as rheumatoid arthritis and psoriasis), which involve the influx of various cells of the host immune system to the inflammation site and release a large number of inflammatory mediators to enhance inflammation. Inflammation is primarily divided into two types: acute inflammation (winds up less than 48 h and neutrophils are main cell type) and chronic inflammation (rests more than 48 h, lymphocytes, macrophages, plasma cells are major participating cells). Acute inflammation may lead to chronic inflammation. Cytokine expression is induced not only by acute inflammation but also by chronic inflammation and is also considered responsible for skin pigmentation. Proinflammatory cytokines (that favor inflammation) or anti-inflammatory cytokines (that neutralize inflammation) accelerate/decelerate the process of inflammation and are involved in controlling inflammatory reactions either indirectly or directly and stimulate/suppress production of cell adhesion molecules or several other inflammatory molecules in diverse cell types.

Generally, pathogen-associated molecular patterns (PAMPs) are derived from microorganisms and trigger the inflammatory response to infections through activation of germline-encoded pattern-recognition receptors (PRRs) [62,63]. Lipopolysaccharide (LPS) is the most common PAMP found on the outer cell wall of Gram-negative bacteria. On the other hand, damage-associated molecular patterns (DAMPs) are derived from host cells exposed to environmental stimuli, such as trauma, heat, and UVR, or from damaged tissue due to ischemia, hypoxia, or genetic defects. DAMPs do not require pathogenic infection but induce sterile inflammatory responses. PRRs families consist of the Toll-like receptors (TLRs), C-type lectin receptors (CLRs), retinoic acid-inducible gene (RIG)-I-like receptors (RLRs), and NOD-like receptors (NLRs) [64]. Among them, TLRs mostly participate in activation of inflammatory responses as a consequence of PAMPs and DAMPs interaction [65]. TLRs activate an intracellular signaling cascade and transfer signals through myeloid differentiation primary response protein-88 (MyD88) leading to nuclear translocation of transcription factors, such as activator protein-1 (AP-1) and nuclear factor kappa-B (NF-κB) or interferon regulatory factor (IRF) 3 (Figure 2). TLRs are shared by both DAMPs and PAMPs to maintain similarities between infectious and noninfectious inflammatory responses [66,67].

Primary inflammatory stimuli including microbial products and cytokines such as IL-1β, IL-6, and tumor necrosis factor (TNF)-α, interact with receptors including TLRs, IL-1 receptor (IL-1R), IL-6 receptor (IL-6R), and the TNF receptor (TNFR), respectively, and trigger important intracellular signaling pathways, including the MAPK, NF-κB and Janus kinase-signal transducer and activator of transcription (JAK-STAT) pathways [68,69,70]. Additionally, it is observed in many inflammatory disorders that ROS have the most significant effects in MAPK/AP-1, NF-kB, and JAK-STAT signaling pathways (Figure 3) [71,72,73,74].

UVR is the most important extrinsic factor in melanogenesis regulation as well as ROS production and is considered to be the main stimulus for induced or acquired pigmentation, known as “tanning” [8,17,19,29]. The main cause of sunburn is UVB irradiation. Skin modulates the immune response to UVB irradiation, leading to neutrophil accumulation in the skin, and mast cells play a crucial role in UVB-induced skin inflammation by inducing a variety of proinflammatory mediators [75,76,77]. Skin UVB exposure significantly increases release of IL-8 and causes a modest increase in release of IL-1; however, other cytokines, such as IL-2, IL-4, IL-6, IL-10, IL-12, IL-13, TNF-α and IFN-γ, remained unchanged. Exposure of skin to both UVA and UVB radiation causes an immediate rise in nicotinamide adenine dinucleotide phosphate (NADPH) oxidase level, resulting in ROS increase which then activates signaling pathways in keratinocytes in the epidermis and fibroblasts in the dermis, leading to the activation of inflammatory genes [78]. UVR can activate signaling pathways linked to surface receptors such as the EGF receptor [79,80], Prostaglandin E2 (PGE2) receptor [81,82], TNF-α receptor [83], and IL-1 receptor [84,85] either by causing receptor clustering [86,87] or by producing ROS, which then promotes phosphorylation and activation of downstream signaling pathway intermediates such as Erk1/2 and p38 [80,88,89,90,91].

Inflammatory mediators are produced after activation of intracellular signaling pathways by inflammatory stimuli. These have an important role in melanogenesis. 

## 5. Inflammatory Cytokines 

Allergens, pathogens, chemical stimuli, physical damage, and UVR can all lead to skin inflammation and inflammatory cytokines are predominantly released from immune cells, including monocytes, macrophages, and lymphocytes, which are considered closely related to skin pigmentation [92,93]. Pro- and anti-inflammatory cytokines facilitate and inhibit inflammation. Cytokines modulate immune response to infection or inflammation and regulate inflammation itself via a complex network of interactions. Th1, Th2 and Th17 cells [94] are the main types of Th cells in inflammation. Th1 cells secret cytokines such as interferon-γ (IFN-γ), TNF, and IL-2 and thus play an important role in cellular immune responses and Th2 cells play a central role in humoral immune responses by secreting IL-4, IL-5, IL-10, and IL-13 as well as other cytokines [95,96]. Th17 cells are a recently recognized subpopulation of T cells and produce IL-17, and IL-22, and IFN-γ. Skin samples from atopic dermatitis (AD) patients contain numerous Th17 and Th2 cells, and cytokines produced from these cells (especially Th2 cells), which reduce the epidermis barrier function, elevate trans-epidermal water loss, and increase permeation of environmental irritants and allergens. Antigen-presenting cells (APCs) in the epidermis and Langerhans cells capture allergens with their dendrites extending upwards and process and present them to lymphocytes in the draining lymph nodes, thus stimulate naïve T-cells to be effector memory T cells. 

Exposure to sensitized allergens evokes activation and accumulation of effector memory T cells to the exposure site, and these cells are stimulated to produce proinflammatory cytokines into the inflamed area. Atopic dermatitis patients are usually sensitized to food allergens, such as egg, and wheat, or to environmental allergens, such as house dust mites and pollen. These allergens existing everywhere can easily provoke inflammation in barrier-disrupted, easily permeable skin of atopic dermatitis patients. Atopic dirty neck is a well-known pigmentary condition considered to be due to post-inflammatory pigmentation/depigmentation. Th17 secretes IL-17, IL-6, IL-21, and IL-22, which participate in innate and acquired immunity. Psoriasis is another major inflammatory skin disease, where Th17 cells play an inevitable role in its pathogenesis. Treatment with biologics often leaves hypopigmented/hyperpigmented macules in the place of psoriasis plaques, which may be reflected by cytokine profiles in the psoriasis plaques. Treatment with UV (Psolalen-UVA (PUVA) or narrowband UVB) leaves strongly hyperpigmented macules without exception, probably due to the melanogenic function of UV.

## 6. Inflammatory Cytokines Induce Melanogenesis

Several studies have observed the involvement of local inflammatory factors in skin pigmentation regulation, which are presented in Table 1 (Figure 4). It has been revealed that IL-4 directly takes part in inhibiting melanogenesis in normal human epidermal melanocytes (NHEMs) by downregulating MITF, TYRP1, and TYRP2 expression through the JAK2-STAT6 signaling pathway [16]. IL-4 is mainly produced by Th2 cells. In chronic inflammation, basophils, eosinophils, and mast cells can also secrete IL-4 [97,98]. IL-4 plays a crucial role in IgE generation in hypersensitivity, inflammation induction, autoimmunity [99] and is also involved in the maintenance of Th2 lymphocytes and acts as an autocrine growth factor of differentiated Th2 cells [100]. It is hypothesized that skewed balance to Th1 is directly responsible for vitiligo development by changing IFN-γ/IL-4, Tbet/Gata3 profiles in vitiligo patients compared to controls [101,102]. 

IL-13 is a typical Th2 cell cytokine also secreted by CD4 cells, natural killer T cells, mast cells, basophils, eosinophils, and group 2 innate lymphoid cells (nuocytes) [103] sharing a receptor of similar structure and JAK2-STAT6 signaling pathway with IL-4 [104,105]. Recently, it has been found that activated human epidermal γδ T cells produce larger amounts of IL-13 than IL-4 [106] and enhance IL-13 production following treatment with Ginsenoside F1 (GF1) extracted from plant genus Panax. This plays a role in the skin-whitening effect of GF1 via effective suppression of tyrosinase and DCT, indicating that IL-13 could be a direct modulator of melanogenesis through the JAK2-STAT6 signaling pathway in skin disorders accompanied by enhanced T-cell responses, such as atopic dermatitis and vitiligo [107].

IL-17 is known to prevent bacterial and fungal infections [108] and can induce the release of large amounts of proinflammatory cytokines from a variety of cells, such as epithelial cells, endothelial cells, and fibroblasts [109], and its effects are vastly increased when cooperating with TNF [110]. IL-17, a proinflammatory cytokine, is produced primarily by Th17 cells, as well as by other immune cells, including neutrophils, natural killer cells, mast cells, αβ and γδT cells, and group 3 innate lymphoid cells. Studies have revealed IL-17 can synergize with TNF to inhibit signaling pathway for melanogenesis, thereby inhibiting pigmentation [93]. For example, studies on psoriasis lesions showed that overexpression of IL-17 and TNF increase melanocyte number and cause a simultaneous decrease in pigmentation signaling. Neutralization of TNF and/or IL-17 with mAbs revealed that both can recover growth and pigment production of melanocytes, which may contribute to pigmentation changes associated with psoriasis treatment [93].
ijms-22-03970-t001_Table 1Table 1Mechanisms of cytokines in the regulation of melanogenesis.CytokinesMain SourceCells Used for ExperimentImpact on MelanogenesisMechanisms of MelanogenesisRefs.IL-4Th-cellsMelanocytesInhibitionJAK2-STAT6 signaling pathway is used for downregulating the expression of MITF, TYRP1, TYRP2[16]IL-13Th2γδ T-cellInhibitionSuppress tyrosinase and DCT through the JAK2–STAT6[107]IL-17Epithelial cells, Endothelial cells,FibroblastsMelanocytes, Primary pooled human keratinocytesInhibitionTNF and IL-17 simultaneously inhibit melanin formation through PKA and MAPK signaling pathways[93]IL-33Keratinocytes, FibroblastsKeratinocytesPromotionActivate p38 and PKA pathways and thus promote MITF, TYR, TYRP1, TYRP2 expression[111,112]IL-18Monocytes/Macrophages, KeratinocytesMelanocytesPromotionIncrease the activity of tyrosinase and upregulate the expression Of TYRP1 and TYRP2[113]IL-1αLangerhans cells, Melanocytes, keratinocytesPrimary melanocytes and swine skinPromotionMelanin deposition is increased when IL-1α is combined with KGF[114]IL-1βMonocytes/Macrophages, KeratinocytesMelanoma cell lines (LB2259-MEL and CP50-MEL)InhibitionThrough the NF-κB and JNK pathways, it downregulates MITF-M expression[115]IL-6Macrophages, T-cells, AdipocyteMelanocytesInhibitionTyrosinase activity is declined[15]PGE2 and PGF2αFibroblasts, KeratinocytesKeratinocytesPromotionActivate cAMP-dependent pathway and stimulate melanocyte dendrite formation[116]INF-γT-cells, NK cells, NKT cellsB16F10InhibitionBlock melanosomal maturation and upregulate STAT1 phosphorylation[117,118]TNFMonocytes, Macrophages, Keratinocytes, Dendritic cells, Th1, Th17 and Th22Melanocytes,Primary pooledhuman keratinocytesPromotionStimulate PKA and MAPK signaling pathways with combination of IL17, and thus inhibit melanin formation[93]GM-CSFMacrophages, Keratinocytes and Th cellsMelanocytesPromotionStimulate MAPK pathway and promote melanocyte proliferation, melanin and synthesis[119]IL-33, interleukin-33; IL-18, interleukin-18; PGE2, prostaglandin E2; PGF2α, prostaglandin F2α; IL-1α, interleukin-1α; GM-CSF, granulocyte-macrophage colony stimulating factor; IL-4, interleukin-4; IL-13, interleukin-13; IL-6, interleukin-6; IL-17, interleukin-17; IL-1β, interleukin-1β; TNF, tumor necrosis factor; IFN-γ, interferons-γ; PKA, protein kinase A; MAPK, mitogen-activated protein kinase; JNK, c-Jun N-terminal kinases; JAK-STAT, Janus kinase-signal transducer and activator of transcription; NF-κβ, nuclear factor kappa-B; TYR, tyrosinase; TRRP1, tyrosinase-related protein-1; TRRP2, tyrosinase-related protein-2; MITF, microphthalmia-associated transcription factor; NK cell, natural killer cell; NKT cell, natural killer T cell; Th, T helper cell.

Recent research has shown that IL-33 can improve melanin biosynthesis in NHEM, stimulate phosphorylation of p38 and CREB, and increase TYR, TYRP1 and TYRP2 (DCT) expression through MITF, resulting in augmentation of melanogenesis [112]. Abundant IL-33 mRNA is found in keratinocytes induced by UVB [111] and fibroblasts [120,121], suggesting injury on skin causes IL-33 release, resulting in skin pigmentation. Initially, IL-33 can induce mast cells to produce proinflammatory cytokines and chemokines (including monocyte chemoattractant protein-1 and macrophage inflammatory protein-1α) [122,123,124,125,126], leading to activation of macrophages [127,128,129], CD 4+T cells, basophils, dendritic cells and neutrophils [122,129,130,131,132], and likely promoting Th2-skewed skin inflammation, which is another indirect effect on pigmentation by IL-33 [133]. It has also been discovered that IL-33 activates and binds its receptor suppressing tumorigenicity 2 (ST2)L/IL-1RAcP on different types of cells, including mast cells and Th2 cells [134]. Then, IL-33 activates NF-κB, suggesting that it regulates outcome of diseases such as atopic dermatitis, which can trigger pigmentation [134]. 

IL-18 may also modulate both innate and adaptive immunity and its dysregulation can cause autoimmune or inflammatory disease, but some studies recently suggest its participation in pigmentation regulation. It has been observed that IL-18 promotes melanogenesis and upregulates TYRP1 and TYRP2 expression [113,118] by activating pathways and increasing cascade expressing MITF [113,118]. These results indicate the involvement of IL-18 in the regulation of pigmentation by regulating melanogenesis.

IL-1 is an important proinflammatory cytokine involved in initiating inflammation and immunological reactions [135], which contributes to increased tumor invasiveness, metastasis, and angiogenesis under chronic inflammatory conditions [115]. IL-1 is produced predominantly from monocytes and macrophages [136], although it is also produced by another types of cells, such as keratinocytes and fibroblasts [137]. IL-1 possesses two forms, IL-1α and IL-1β [138]. Although they share only 24% identity in protein sequence, IL-1β and IL-1α fold in an extremely similar manner and bind to the same receptor—the type I and type II IL-1 receptor (IL-1RI, IL-1RII, respectively) [139]. The expression patterns of IL-1α and IL-1β vary among cell types [140]. IL-1α is predominantly produced by keratinocytes. Keratinocytes secrete both IL-1α and IL-1β molecules in vitro [137], in addition to the expression of IL-1RI, IL-1RII [141]. The signal transduction is initiated by binding to IL-1RI [142] which can inhibit tyrosinase activity and melanogenesis [15,115] and also stimulates human fibroblasts to produce keratinocyte growth factor (KGF) [143]. IL-1α released from keratinocytes has been shown to regulate the homeostasis of mammalian skin [144] and sufficient to trigger skin inflammation in humans [141,144] and produce more IL-1α upon UVB exposure [145]. It is documented that KGF-induced TYR expression in primary melanocytes [114]. The combination of KGF and IL-1α increases melanin deposition and is responsible for initial stage of human solar lentigines [114]. On the other hand, it was observed that the expression of MITF-M was inhibited after IL-1β treatment to a panel of melanoma cell lines. The inhibitory effects of IL-1β on melanogenesis were removed by inactivation of NF-κB and JNK pathways, suggesting IL-1β could inhibit melanogenesis by downregulating MITF-M through NF-kB and JNK pathways [115].

IL-6 is produced by numerous different cell types including keratinocytes, fibroblasts, and dermal endothelial cells and plays a critical role in regulating acute phase response, hematopoiesis, inflammation, metabolic control, liver regeneration, bone metabolism and cancer progression by promoting cell growth, survival, and differentiation [146]. It has been revealed in recent studies that the IL-6 cytokine elicits a dose-dependent decrease in tyrosinase activity, thus inhibiting melanocyte proliferation and melanogenesis [15]. 

PGE2 is the most abundant prostaglandin released by keratinocytes in response to UVR and inflammatory conditions such as wound healing and stimulates the formation of dendrites in melanocytes [147,148]. A wide range of physiological processes in the skin, including immune function, carcinogenesis, cutaneous barrier function, cell growth, and differentiation are regulated by PGE2 [149,150,151,152]. Biosynthesis of PGE2 is controlled by phospholipase A2 (PLA2), cyclooxygenase (COX), and prostaglandin E synthase (PGES) enzymes. PGF2α is produced by fibroblasts and keratinocytes and stimulates melanocyte dendrite formation and tyrosinase activation, but not proliferation [151]. PGE2 binds to four distinct G-protein coupled receptors (EP1–4) and EP4 receptor signaling stimulates cAMP production in melanocytes. EP3 receptor-lowered basal cAMP levels suggest relative levels of activity of these receptors’ control effects of PGE2 on cAMP in melanocytes that stimulates tyrosinase activity and proliferation [116,151]. 

IFN-γ is also a common secretory cytokine in the skin and the most important endogenous mediator of immunity and inflammation [118]. Th1 lymphocytes, CD 8+ cytotoxic T lymphocytes and NK cells mainly release the proinflammatory cytokine IFN-γ [153]. However, IFN-γ is also secreted by other cells, including antigen-presenting cells, B cells and NKT cells [154,155,156]. A clear relationship between increased IFN-γ signaling and a decreased number of stage III and IV melanosomes in the lesional skin of leprosy patients have been shown [117]. Specific IRF1 binding sites are in the promoter region of pigmentation-related genes such as TYR, TYRP2 (DCT). It is shown that IFN-γ negatively regulates the expression of DCT and other pigmentation-related genes, providing strong evidence that IFN-γ had a significant role in downregulating melanosome maturation [117] —i.e., skin hypopigmentation [117]. Two other crucial mediators of skin pigmentation, α-MSH and TGF-β, are involved in the protective tanning response and maintenance of melanocytes in an immature state [157,158] and are released during inflammation or UV exposure. Thus, Natarajan VT et al. proposed an αβγ-cytokine regulatory model of melanogenesis [117]. Recent studies in various mouse models of vitiligo have demonstrated that local IFN-γ accumulation produced by melanocyte-specific CD 8+ T cells plays an important role in skin depigmented spots [159,160] and also revealed increased IFN-γ is essential for vitiligo pathogenesis by inducing apoptosis of melanocytes [161]. Moreover, IFN-γ can upregulate STAT1 phosphorylation, and JAK1 inhibitors can suppress inhibiting effect of IFN-γ, thus contributing to melanogenesis. Other studies have also shown that IFN-γ inhibits IL-18-induced melanogenesis [118]. 

TNF functions by binding to two different receptors, TNFR1/p55 and TNFR2/p75 [92], and is secreted mainly by monocytes and macrophages, and also by keratinocytes, dendritic cells, Th1, Th17 and Th22. TNF induces inflammation through activation of vascular endothelial cells and immune cells and plays a crucial role as a regulator of lymphoid tissue development by controlling apoptosis [92]. In several autoimmune diseases, increased levels of TNF have been found at inflammation sites, such as in the lesional skin of psoriasis patients [162]. Overexpression of IL-17 and TNF synergistically modulate cytokine expression, increase melanocyte number while suppressing pigmentation signaling [93]. Therapeutic neutralization of TNF and IL-17 with mAbs resulted in a rapid recovery of pigmentation-related gene expression in psoriasis lesions, indicating IL-17 and TNF can affect pigment production in melanocytes, which may contribute to the hypopigmentation associated with psoriasis. For example, after 24–48h treatment with both IL-17 and TNF of melanocytes, decreased levels of c-KIT, MC1R, MITF, and TYRP2 were observed [93]. Consequently, the levels of tyrosinase and melanin were significantly reduced [93]. It has been suggested the combination of IL-17 and TNF can inhibit melanogenesis through PKA and MAPK signaling pathways [92,93] and blocking TNF can lead to rapid restoration of pigmentation gene expression in psoriatic lesions, indicating that anti-TNF has potential for treating depigmented dermatosis [93]. IL-33 expression is induced by IL-17 and IFN-γ in epidermal keratinocytes and induces melanogenesis, which creates a negative feedback loop in epidermal pigmentation [163,164].

GM-CSF may be a vital factor in the melanogenesis of the skin because of its role in promoting melanocyte proliferation and melanin synthesis. GM-CSF is produced by mononuclear macrophages, keratinocytes and Th cells [119]. It has been found GM-CSF synthesis and secretion by HaCaT cells increased cell proliferation in the wound healing process and contributes to wound healing and melanogenesis at the site of injury [165]. In addition, to predict the prognosis of transplantation of cultured autologous melanocytes (TCAM) in vitiligo patients, increased serum levels of GM-CSF could be used as the serum biomarkers [166].

## 7. Conclusions

From the currently available literature, little is known about the relationship between inflammation and pigmentation. We have attempted to show how diversified stimuli induce skin inflammation and how inflammatory pathways and inflammatory cytokines interact with each other, resulting in skin pigmentation. Our review has demonstrated that several inflammatory factors can promote or inhibit the melanogenesis in melanocytes through different mechanisms. It is widely accepted that the involvement of all types of inflammatory cells in the activation and release of inflammatory mediators makes it more complex to understand the regulatory network of inflammation and pigmentation. We believe that our review elucidating the role of inflammatory factors in the regulation of melanogenesis can provide new therapeutic perspectives for treating pigmentary disorders.

## Figures and Tables

**Figure 1 ijms-22-03970-f001:**
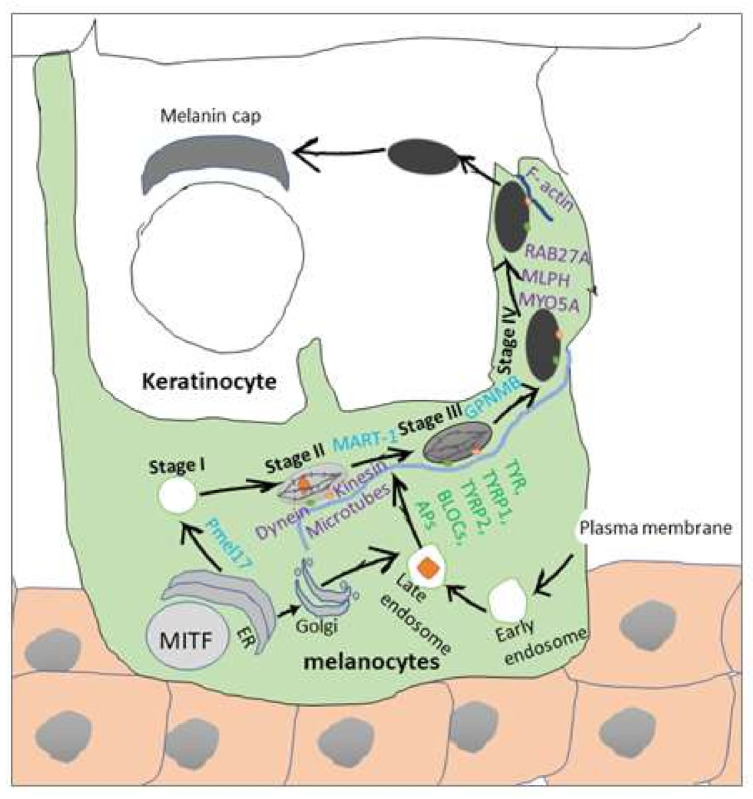
Factors that regulate melanosome formation and maturation through four distinct stages (I–IV) during the melanin production by a melanocyte. The melanogenesis takes place in a special organelles called melanosomes. Melanosomal structural proteins (PMEL17, MART-1, and GPNMB) affect melanosome structure. Enzymes (TYR, TYRP1, TYRP2, BLOCs, APs) are involved in melanin synthesis either direct or indirect manner. Proteins (microtubes, F-actin, Kinesin, dynein, MLPH, MYO5A, RAB27A) ensure intracellular transport of melanosomes. Melanocyte-specific transcription factors (specially MITF) regulate the expression and function of many pigment-specific factors. ER: endoplasmic reticulum; PMEL17: melanocytes lineage-specific antigen gp100; MART-1: melanoma antigen recognized by T cell-1; GPNMB: glycoprotein nonmetastatic melanoma protein b; TYR: tyrosinase; TYRP1: tyrosinase-related protein-1; TYRP2: tyrosinase-related protein-2; BLOCs: biogenesis of lysosome-related organelles complexes; Aps: adaptor protein complexes; MLPH: melanophilin; MYO5A: myosin-Va; RAB27A:Ras-related protein Rab27A; MITF: microphthalmia-associated transcription factor.

**Figure 2 ijms-22-03970-f002:**
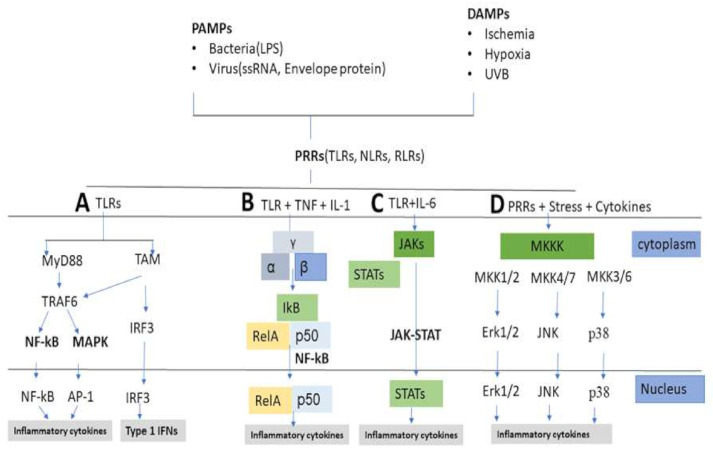
DAMPs and PAMPs bind to PRRs (TLRs, NLRs, RLPs) and activate several signaling pathways. (**A**) TLRs activate MyD88 or TAM signaling pathways, which lead to nuclear translocation of NF-κβ and AP-1 or IRF3, which plays a crucial role in the regulation of inflammatory response. (**B**) TLRs and some inflammatory cytokines, such as IL-1 and TNF trigger NF-κB pathway resulting activation of RelA/p50 complexes that regulate expression of inflammatory cytokines. IKK subunits phosphorylate IκB and regulate NF-κB pathway activation. (**C**) The binding of IL-6 family members to plasma membrane receptors activates the JAKs. JAKs activate STATs which are dephosphorylated in the nucleus and activate downstream cytokines. (**D**) MAPKs direct cellular response to a variety of stimuli, including stress, mitogens and inflammatory cytokines. MKKKs phosphorylate and activate MKKs. Then, MKKs phosphorylate and activate MAPKs including Erk1/2, JNK, and p38. Activated MAPKs phosphorylate various proteins and take part in regulation of inflammatory responses. DAMPs: damage-associated molecular; PAMPs: pathogen-associated molecular patterns; PRRs: pattern-recognition receptors; TLRs: Toll-like receptors; NLRs: NOD-like receptors; RLRs: retinoic acid-inducible gene-I-like receptors; MyD88: myeloid differentiation primary response protein-88; TAM: Tyro3, Axl, and Mer receptors; NF-κβ: nuclear factor kappa-B; AP-1: activator protein-1; IRF3: interferon regulatory factor3; IL-1: interleukin-1; TNF: tumor necrosis factor; IKK: IκB kinase; JAK-STAT: Janus kinase-signal transducer and activator of transcription; IL-6: interleukin-6; MAPKs: mitogen-activated protein kinases; MKKK: MAPK kinase kinase; Erk1/2: extracellular signal-regulated kinase 1/2; MKK: MAPK kinase; JNK: c-Jun N-terminal kinases.

**Figure 3 ijms-22-03970-f003:**
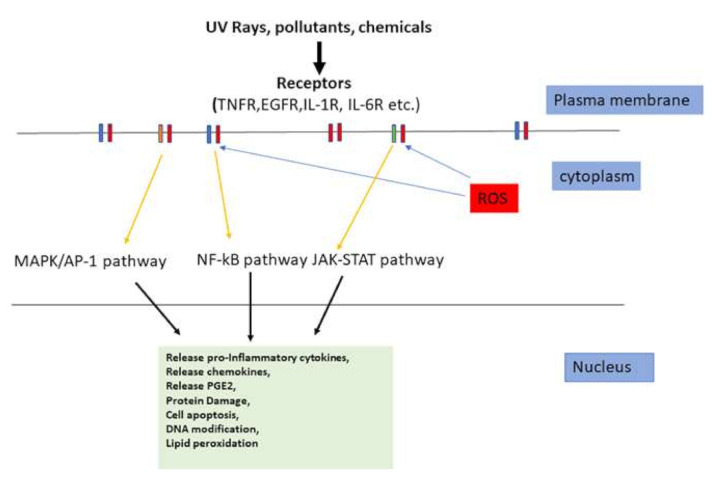
Ultraviolet radiation (UVR), pollutants, and chemicals can activate surface receptors either directly or indirectly by activating reactive oxygen species (ROS). Thus, signaling pathways are activated, resulting in protein damage, cell apoptosis, DNA modification, lipid peroxidation, and production of proinflammatory mediators. UVR: ultraviolet radiation; NF-κβ: nuclear factor kappa-B; AP-1: activator protein-1; IL-1R: interleukin-1 receptor; TNFR: tumor necrosis factor receptor; JAK-STAT: Janus kinase-signal transducer and activator of transcription; IL-6R: interleukin-6 receptor; MAPK: mitogen-activated protein kinase; EGFR: epidermal growth factor receptor; PGE2: Prostaglandin E2. Yellow, blue and black arrows indicate direction.

**Figure 4 ijms-22-03970-f004:**
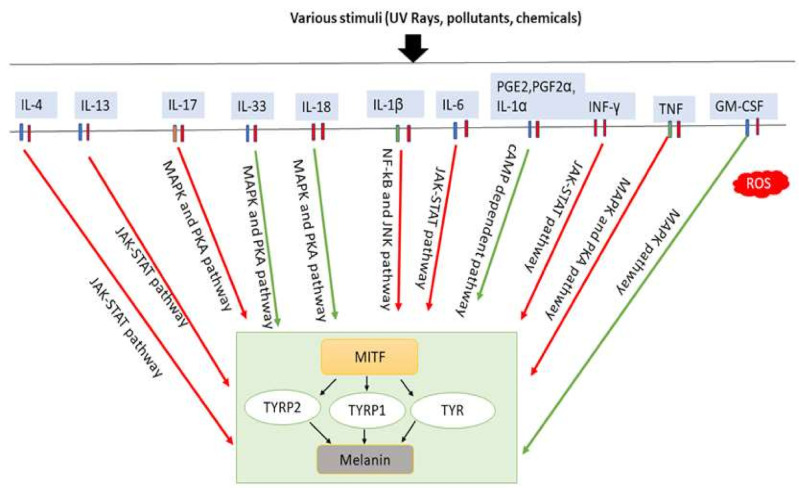
Involvement of inflammatory factors in melanogenesis. Inflammatory factors including IL-33, IL-18, and GM-CSF promote melanogenesis by stimulating the MAPK and PKA pathways either simultaneously or individually. In addition, PGE2, PGF2α, and IL-1α promote melanogenesis by stimulating cAMP-dependent pathway. In contrast, inflammatory factors such as IL-4, IL-13, IL-6, IL-17, IL-1β, TNF and IFN-γ inhibit melanogenesis by suppressing the PKA and MAPK, or JAK-STAT, or NF-κβ and JNK pathways. IL-33: interleukin-33; IL-18: interleukin-18; PGE2: prostaglandin E2; PGF2α: prostaglandin F2α; IL-1α: interleukin-1α; GM-CSF: granulocyte-macrophage colony stimulating factor; IL-4: interleukin-4; IL-13: interleukin-13; IL-6: interleukin-6; IL-17: interleukin-17; IL-1β: interleukin-1β; TNF: tumor necrosis factor; IFN-γ: interferons-γ; PKA: protein kinase A; MAPK: mitogen-activated protein kinase; JNK: c-Jun N-terminal kinases; JAK-STAT: Janus kinase-signal transducer and activator of transcription; NF-κβ: nuclear factor kappa-B; TYR: tyrosinase; TRRP1: tyrosinase-related protein-1; TRRP2: tyrosinase-related protein-2. Red arrows indicates inhibition; green arrows indicate promotion.

## Data Availability

The data presented in this study are available on request from the corresponding author.

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
