# Peer review of "Diversified Stimuli-Induced Inflammatory Pathways Cause Skin Pigmentation"

_ijms, 2021, doi:10.3390/ijms22083970_

Round 1

Reviewer 1 Report

This is an excellent review paper, summarizing the recent finding of the role of inflammatory cytokines on skin pigmentation. The paper is interesting, informative and written on high scientific standard. The main mechanisms of melanogenesis, including melanosomes formation, their transportation, regulatory mechanisms and involved signaling pathways are discussed in details. The recent evidences for the role of inflammatory cytokines in melanogenesis process by activating or suppressing specific signaling pathway have been summarized and the effects of each one on skin pigmentation were suggested. The importance of external and intrinsic stimuli, which can induce acute or chronic skin inflammation, were pointed and their molecular mechanisms were discussed. I would like to recommend the authors to improve the quality of Figure 1. by increasing the size of the text inside to become readable.  Also I have some technical remarks, which do not impact the scientific quality of the manuscript:

  1. 6 - Double check the initials of the authors on title page (K.K.) -?; (M.K.) - ?;
  2. 321 – “…stimulate naïve…”;
  3. 471-472 – All coauthors should be listed with their activities in “Author Contributions”

Author Response

Response to Reviewer 1 Comments

This is an excellent review paper, summarizing the recent finding of the role of inflammatory cytokines on skin pigmentation. The paper is interesting, informative and written on high scientific standard. The main mechanisms of melanogenesis, including melanosomes formation, their transportation, regulatory mechanisms and involved signalling pathways are discussed in details. The recent evidences for the role of inflammatory cytokines in melanogenesis process by activating or suppressing specific signalling pathway have been summarized and the effects of each one on skin pigmentation were suggested. The importance of external and intrinsic stimuli, which can induce acute or chronic skin inflammation, were pointed and their molecular mechanisms were discussed. I would like to recommend the authors to improve the quality of Figure 1. by increasing the size of the text inside to become readable.  Also, I have some technical remarks, which do not impact the scientific quality of the manuscript:

We thank the reviewer for the thoughtful review and crucial comments which have helped guide the revision of the manuscript. We have increased the size of text inside of figure 1, as you recommended. Our replies are as follows:

Point 1:  1.6 - Double check the initials of the authors on title page (K.K.) -?; (M.K.) - ?;

Response 1: We apologize for this unintentional mistake. We have made a correction in the initials of the authors on title page (page 1, line 6).

1Department of Dermatology, Faculty of Medicine, Department of Dermatology, Jichi Medical University, Tochigi, Japan; [email protected] (M.R.H.); [email protected] (T.M.A.); [email protected] (M.O.); [email protected] (M.K.)”

Point 2: 321 – “…stimulate naïve…”;

Response 2: We would like to appreciate the reviewer`s suggestion regarding the correct word choice. We have modified the sentence according to the suggestion (page 9, lines 321).

“Antigen-presenting cells (APC) in the epidermis and Langerhans’s cells capture allergens with their dendrites extending upwards, process and present them to lymphocytes in the draining lymph nodes, and thus stimulated naïve T-cells to be effector memory T cells.”

Point 3: 471-472 – All coauthors should be listed with their activities in “Author Contributions”.

Response 3: We would like to appreciate the reviewer`s suggestion regarding all co-author’s activities in “Author Contributions” section. We have specified all co-authors activities according to the reviewer’s comment (page 3, lines 471-472).

Reviewer 2 Report

The topic addressed by the authors is really current and relevant.
Although the originality of the topic is not high, given the high number of articles about this issue, the authors have conducted a good review of the scientific literature.
The manuscript is correctly written and easy to read. The conclusions are consistent with the arguments presented.

Author Response

Response to Reviewer 2 Comments

The topic addressed by the authors is really current and relevant.
Although the originality of the topic is not high, given the high number of articles about this issue, the authors have conducted a good review of the scientific literature.
The manuscript is correctly written and easy to read. The conclusions are consistent with the arguments presented.

We thank the reviewer for the judicious review and encouraging remarks which have enriched our thoughtfulness and guided us to the revision of the manuscript.

List of changes made in the revised manuscript-

  • The entire manuscript has been carefully checked; English and errors have been corrected.
  • We have increased the size of the text inside of figure 1 (page 4).
  • Text was changed in the manuscript ( line 6) , ( line 321) (line 471-472) .
